# Development of a scale for the evaluation of the quality of the shared decision process in multiple sclerosis patients

Elena Álvarez-Rodríguez[1], César Manuel Sánchez-Franco[2], María José Pérez-Haro[3], Laura Bello-Otero[1], Marta Aguado-Valcarcel[1], Inés González-Suárez[1]*

1 Neurologist, Neurology Department, Hospital Álvaro Cunqueiro, Vigo, Spain, 2 School of Nursing, Neurology Department, Hospital Álvaro Cunqueiro, Vigo, Spain, 3 Biostatech, Advice, Training and Innovation in Biostatistics, S.L. Santiago de Compostela, Spain

* igonsua@gmail.com

**Citation:** Álvarez-Rodríguez E, Sánchez-Franco CM, Pérez-Haro MJ, Bello-Otero L, Aguado-Valcarcel M, González-Suárez I (2022) Development of a scale for the evaluation of the quality of the shared decision process in multiple sclerosis patients. PLoS ONE 17(5): e0268125. https://doi.org/10.1371/journal.pone.0268125

**Data Availability Statement:** All relevant data are within the paper and its Supporting information files.

## Abstract

In the last years, therapeutic decisions in multiple sclerosis (MS) have become challenging due to expanded options with different treatment profiles attending to efficacy, safety, and route and frequency of administration. Moreover, patients with multiple sclerosis (PwMS) increasingly wish to be involved in their therapeutic decision process. Therefore, a new, patient-centric shared decision model (SDM), is gaining relevance. However, validated scales oriented to assess the quality of the process itself are lacking. The AGA-25 scale is a fit-for-purpose 25-item scale based on two validated scales in MS (Treatment Satisfaction Questionnaire for Medication (TSQM) and Decisional Conflict Scale (DCS)). The aim of this work is to develop and validate the AGAS-25 in Spanish. Two hundred and three PwMS (aged 17 to 67; 155 [76.4%] females) undergoing stable disease modifying treatment in the last 6 months were consecutively recruited. The Principal Component Analysis suggested a four-factor structure for the 25-item version of the questionnaire: 1) satisfaction with the SDM process 2) adverse events with the DMT, 3) convenience of the chosen-DMT and 4) information reliability. The internal consistency of the measurement was adequate (Cronbach's alpha = 0.88). Our results support the use of the AGAS-25 scale to assist SDM in Spanish-speaking PwMS.

## Introduction

Multiple sclerosis (MS) is a chronic neurodegenerative disorder characterized by inflammation and progressive neurological destruction and degeneration [1]. Although there is no cure for MS, several disease modifying treatments (DMTs) have demonstrated to be effective by reducing the frequency of clinical relapses and disability progression. Nowadays, more than 15 different DMTs are available, the approved DMT landscape includes drugs with different therapeutic profiles based on their efficacy, safety, and route and frequency of administration, which impact patient preference and adherence [2].

**Funding:** The authors received no specific funding for this work.

**Competing interests:** The authors have declared that no competing interests exist.

Patients' satisfaction is recognized as an important dimension of the quality of care since it has been related to patient compliance, doctor-patient information exchange and continuity of care. An important dimension of patient satisfaction is shared decision making (SDM) [3]. SDM is defined as an approach where clinicians and patients share the best available evidence when faced with the task of making decisions, and where patients are supported to consider options and to achieve informed preferences [4]. Therefore, the exchange of information is the central axis of the process.

In this exchange, both parts add their experience, ethos, and preferences to the final decision [5]. SDM recognizes that both the patient and the clinician shared different but equally valid experiences and expertise to the decision making process. At best, clinicians add their knowledge of the treatments, their outcomes and prognosis, whether the patient displays how the disease impacts their life, their personal values and risk tolerance [6]. SDM is recommended in the majority of healthcare decisions where there is more than one feasible option and is particularly suited for a chronic condition such as MS, improving patients' perceptions of the benefits and risk of the different DMT [7].

Although the most widespread practice is explicit information, several approaches have been investigated. While there is currently no 'gold standard' in terms of measurement, the need to measure the process as well as the outcome is apparent [6]. Only a few scales are available to assess SDM process from the patients satisfaction point of view [8]. The 9-item Shared Decision Making Questionnaire (SDM-Q-9) [9], have revealed poor quality of evidence, suggesting that its value as an assessment tool may be limited [10]. Prior studies evaluating multiple sclerosis patients (PwMS) understandings and preferences have been developed [7].

A recent review demonstrated that the experience of many patients with information during the standard process does not provide satisfactory understanding of the risks and benefits of DMTs. It is known that PwMS tend to underestimate the risks associated with DMTs, which could lead to lower adherence rates and greater discontinuation due to adverse events [6]. However, some studies have observed an overestimated point of view of the DMTs that can also have an impact on adherence [11].

Therefore, the evaluation of the SDM process should not only assess the PwMS degree of involvement in making decisions but also whether the information received was consistent with their subsequent personal experience in terms of risk, efficacy, and satisfaction [12, 13].

Two of the most used scales in clinical practice are the Treatment Satisfaction Questionnaire for Medication (TSQM) and the Decisional Conflict Scale (DCS), both were designed as global scales. TSQM is used as a general measure of treatment satisfaction with medication. TSQM has been validated in several diseases, including MS [14–16]. The scale evaluates the patient's drug satisfaction in terms of effectiveness, side effects, convenience, and global satisfaction. The 16-item decisional conflict scale was developed to elicit information concerning the decision makers: 1) uncertainty in making a choice; 2) modifiable factors contributing to the uncertainty, such as lack of information, unclear values, and inadequate social support; and 3) perceived effective decision making [17]. Based on these scales we developed a 25-item scale to analyze the quality of the SDM in a MS outpatient clinic. The main aim of this work is to assess the psychometric properties of the self-administered questionnaire AGA-25 developed in Spanish to ensure the quality of the SDM process in MSPw.

## Materials and methods

A written informed consent was obtained from each participant. The Clinical Research Ethics Committee of Galicia gave its approval to the study with a research code 2018/271. The investigations were consistent with the principles outlined in the Declaration of Helsinki.

## Questionnaire development

The team used the Medline database to search prior studies. The keywords "shared-decision process", "satisfaction questionnaire", and "multiple sclerosis" were considered, and relevant information was classified and discussed afterwards [6, 7].

## Formulation of the AGA questionnaire

1. The Treatment Satisfaction Questionnaire for Medication (TSQM) comprises 14 items across four domains focusing on effectiveness (three items), side effects (five items), convenience (three items), and global satisfaction (three items) of the medication. TSQM was designed to assess patient treatment satisfaction in chronic diseases and has been validated into Spanish [18] and used in several diseases, including MS [14–16]. This scale was developed for the analysis of satisfaction with oral medications. However, some authors believed that satisfaction measures are often misleading, as high satisfaction scores are more likely to be the result of low expectations than a high quality SDM process [18].

2. The decisional conflict scale (DCS) is a measure of the uncertainty surrounding a treatment choice and patient confidence in making that decision; it also evaluates modifiable factors contributing to the uncertainty, such as lack of information, unclear values, and inadequate social support and perceived effective decision making [17]. However, it is not a measure of the quality itself.

Based in the Weaber conceptual Framework for Treatment Satisfaction [20]; three experienced physicians selected the most reliable items and developed a 25-items scale (see Table 1) looking for 4 different dimensions: 1) satisfaction with the information during the SDM (items 01, 02, 03, 04, 05, 06, 11, 12 and 25), exploring whether the doctor involved the patient in the decision, offered detailed information, answered questions or had to search information from other sources (INTERNET, patients, associations); 2) the adverse events during the selected treatment (items 15, 16, 17, 18 and 19), presence of adverse events with DMT and the interference in his/her day-to-day life; 3) DMT convenience (items 7,8,10,20 and 21), if the chosen DMT fits into the PwMS rhythm of life and 4) information reliability (items 13, 14, 22, 23 and 24), how satisfied the patient is with the current treatment and his/her confidence in him/herself to control the medication. The answers ranged from totally disagree, disagree, agree, to totally agree (1,2,3,4).

## Data collection

One hundred and forty-two PwMS were randomly selected from the MS outpatient clinic. The study was conducted during 2019 at the Hospital Álvaro Cunqueiro, a public hospital in Vigo, Spain. Inclusion criteria were 1) MS diagnosis using McDonald criteria 2010; 2) stable treatment in the last 6 months; 3) absence of relapses in the past 6 months; 4) capacity to sign an informed consent. Demographic data were collected including age and sex, DMT and time on DMT, type of MS, age at onset of the first symptom and to diagnosis, EDSS, TAB and total number of relapses, prior DMT and reason for the switch.

Patients understood the goal of the study and received the printed questionnaire. The survey was completed anonymously.

## Statistical analysis

A descriptive analysis has been carried out. Quantitative features are shown through the mean (SD), median and range (minimum and maximum). On the other hand, qualitative variables are described by absolute and relative (%) frequencies.

**Table 1. Questionnaire design.**

| | |
|---|---|
| Item 01 | Se me explicó detalladamente la razón por la que era necesario iniciar un tratamiento para la esclerosis múltiple. |
| | The doctor explained me in detail why it was necessary to start a treatment for MS. |
| Item 02 | Se me consultó cómo de implicado me gustaría estar a la hora de tomar decisiones con respecto al nuevo tratamiento. |
| | I was asked how involved I would like to be in making decisions about my new treatment. |
| Item 03 | Se me explicó que para mí esclerosis múltiple había diferentes opciones de tratamiento. |
| | The doctor explained me, that there were multiple sclerosis treatment options. |
| Item 04 | Se me informó de manera detallada de las ventajas de cada uno de los tratamientos para la esclerosis múltiple. |
| | I was informed in detail of the benefits of each of the MS treatments. |
| Item 05 | Se me resolvieron las dudas de manera que pude entender la información. |
| | If I had doubts, they were resolved in a way that I could understand all the information. |
| Item 06 | ¿En qué grado entendió en qué consistía el tratamiento elegido? |
| | To what degree, did you understand the chosen treatment? |
| Item 07 | ¿En qué grado entendió los efectos secundarios relacionados con la 1ª dosis? |
| | To what degree, did you understand the side effects related to the first dose? |
| Item 08 | ¿En qué grado entendió los efectos secundarios a largo plazo? |
| | ? To what degree, did you understand the long-term side effects? |
| Item 09 | ¿Cómo percibe el riesgo de presentar un evento adverso a lo largo del tratamiento? |
| | How do you perceive the risk of experiencing an adverse event throughout the treatment? |
| Item 10 | En algún momento del proceso de toma de decisiones tuve que buscar información en otros medios. |
| | During the decision-making process, I had to look for information in other media. |
| Item 11 | La decisión fue tomada de manera conjunta entre el especialista y yo. |
| | The decision was made jointly by the specialist and me. |
| Item 12 | Tras decidir el tratamiento adecuado, decidimos el modo de proceder adecuado. |
| | After deciding on the appropriate treatment, we decided the best procedure. |
| Item 13 | ¿En qué grado cree que el medicamento es capaz de prevenir un brote de su enfermedad? |
| | To what degree, do you think the medicine is able to prevent an outbreak of your disease? |
| Item 14 | ¿En qué grado cree que el medicamento es capaz de prevenir la progresión de su enfermedad? |
| | ? To what degree, do you think the medicine is able to prevent the progression of your disease? |
| Item 15 | ¿Padece efectos secundarios a consecuencia del medicamento? |
| | Do you have any side effects from this medicine? |
| Item 16 | ¿En qué grado le molestan esos efectos secundarios en su día a día? |
| | To what degree, do these side effects bother you in your day-to-day life? |
| Item 17 | ¿Hasta qué punto interfieren esos efectos secundarios en su salud física? |
| | ? To what degree, do these side effects interfere with your physical health? |
| Item 18 | ¿Hasta qué punto interfieren esos efectos secundarios en su salud emocional? |
| | To what degree, do these side effects interfere with your emotional health? |
| Item 19 | ¿Hasta qué punto influyen estos efectos secundarios en su satisfacción con el medicamento? |
| | To what degree, do these side effects influence your satisfaction with the medication? |
| Item 20 | La forma de administración del medicamento le parece sencillo. |
| | How simple do you find the way to administer the medicine? |
| Item 21 | La planificación de la toma del medicamento le parece sencilla. |
| | How simple is it to plan the shot? |
| Item 22 | ¿Qué percepción tiene de la eficacia del medicamento a la hora de controlar su enfermedad? |
| | What is your perception of the effectiveness of the drug in controlling your disease? |

(*Continued*)

**Table 1.** (Continued)

| Item 23 | Teniendo en cuenta ventajas e inconvenientes del fármaco, ¿cómo está de satisfecho con el mismo? |
|---|---|
| | Considering the advantages and disadvantages of the drug, how satisfied are you with it? |
| Item 24 | Las ventajas superan a las desventajas del fármaco. |
| | The advantages outweigh the disadvantages of the drug. |
| Item 25 | La información recibida durante el proceso de toma de decisiones ha sido acorde a la experiencia con el fármaco. |
| | The information received during the decision-making process has been consistent with my experience with this drug. |

The model was built with all the data, extracting the number of factors through a principal components analysis (PCA). Before performing PCA, Kaiser-Meyer-Olkin (KMO) index and Bartlett's sphericity test were calculated to analyze its efficiency. The internal consistency of items and the reliability of each dimension were assessed through the Cronbach's alpha reliability coefficient.

Statistical analyzes have been carried out with free software R (R Core Team 2020). The significance level was set up at 0.05.

## Results

### Descriptive analysis

A summary of the demographic and clinical features is shown in Table 2. Most of the patients (76.4%) were women with a mean age of 41.32 years (SD 8.73). Most of the patients were on stable first-line therapy (61.1% vs 38.9%). Patients were mildly disabled with a mean EDSS score of 2.14 (SD 1.76) and a mean disease duration of 8.80 years (SD 6.34).

### Item selection

The consistency of the scale was analyzed through Cronbach's alpha reliability coefficient. For the global questionnaire, the alpha was 0.88 and it did not improve when any of the items were deleted. However, when the item-total correlation was assessed, low item-total correlations show that that item doesn't correlate well with the scale overall. Item 9 showed the worst correlation in comparison to the other items (Table 3) and was therefore deleted from the final questionnaire.

### Principal component analysis

Principal component analysis proved to be a strong mechanism to factorize the data via Bartlett's sphericity test ($p < 0.001$) and the Kaiser-Meyer-Olkin index (KMO = 0.85).

To guarantee the best structure, Horn's Parallel analysis (PA) was performed. This methodology [21] compares the eigenvalues of the original dataset to the eigenvalues from other randomly generated same-size data. All the principal components associated with eigenvalues lower than those from the generated dataset will be excluded. A graphical representation of the PA outcome is shown in Fig 1. According to this exploration, there are 4 dimensions to retain.

In addition, the Kaiser's rule [22] was checked, and conforming to the previous outcome, 4 dimensions should be selected as well. These four eigenvalues are 6.2, 3.3, 2.0, and 1.1, and they explain 28.8%, 19.7%, 9.1%, and 6.4% of the variance. That is, 64.2% of the total variance.

**Table 2. Descriptive analysis.**

| Gender | |
|---|---|
| Men | 48 (23.6%) |
| Women | 155 (76.4%) |
| **DMT** | |
| Alemtuzumab | 25 (12.3%) |
| Interferon beta-1a | 10 (4.9%) |
| Interferon beta-1b | 2 (1.0%) |
| Cladribina | 2 (1.0%) |
| Glatiramer acetate | 19 (9.4%) |
| Dimetilfumarate | 31 (15.3%) |
| Fingolimod | 27 (13.3%) |
| Natalizumab | 21 (10.3%) |
| Ocrelizumab | 3 (1.5%) |
| Sc interferon beta-1a | 31 (15.3%) |
| Rituximab | 1 (0.5%) |
| Teriflunomide | 31 (15.3%) |
| **Age at first symptom** | |
| Mean (SD) | 30.025 (8.901) |
| Median | 29.715 |
| Range (Min-Max) | 10.564–59.767 |
| **Years from diagnosis** | |
| Mean (SD) | 8.803 (6.335) |
| Median | 7.381 |
| Range (Min-Max) | 0.367–24.400 |
| **Age** | |
| Mean (SD) | 41.324 (8.726) |
| Median | 41.142 |
| Range (Min-Max) | 17–67 |
| **EDSS** | |
| Mean (SD) | 2.140 (1.755) |
| Median | 2.000 |
| Range (Min-Max) | 0.000–7.000 |

The loading factors settle the weight of each item in each dimension. A varimax rotation has been performed to simplify the results. All the outputs have been collected in (Table 4). The calculated Cronbach's alpha reliability coefficient for each dimension was within the range from 0.63 to 0.93, showing an acceptable internal consistency for all the cases.

## Discussion

We created the first questionnaire to assess the quality of the information administered during the SDM process in MS patients. Our study demonstrated that AGA-25 scale is a feasible, reliable and valid questionnaire for use in clinical practice with patients with MS.

In recent years, a focused-centered approach in which the patient is the core of the healthcare system has been increasingly recognized. Year by year MS drugs are increasing. The increasingly approved DMT landscape includes drugs with different profiles in terms of routes of administration (injectable, oral, and infusion), frequencies, mechanisms of action, and safety and tolerability profiles. In this increasingly complex scenario, it seems vital to include

**Table 3. Item-total correlation without the item itself.**

|  | Item-Total Correlation |
|---|---|
| Item 01 | 0.41 |
| Item 02 | 0.48 |
| Item 03 | 0.56 |
| Item 04 | 0.57 |
| Item 05 | 0.50 |
| Item 06 | 0.54 |
| Item 07 | 0.43 |
| Item 08 | 0.50 |
| Item 09 | 0.17 |
| Item 10 | 0.32 |
| Item 11 | 0.50 |
| Item 12 | 0.48 |
| Item 13 | 0.34 |
| Item 14 | 0.42 |
| Item 15 | 0.46 |
| Item 16 | 0.49 |
| Item 17 | 0.42 |
| Item 18 | 0.44 |
| Item 19 | 0.39 |
| Item 20 | 0.35 |
| Item 21 | 0.35 |
| Item 22 | 0.50 |
| Item 23 | 0.54 |
| Item 24 | 0.56 |
| Item 25 | 0.58 |

the patient's preferences and values at the center of the decision. This kind of decisions have been called, preference-sensitive and they reflect the fact that the medical evidence is necessary, but not sufficient. A new, patient-centric shared decision model (SDM), is gaining relevance. SDM has been associated with higher adherence rates [19]. Thus, is increasingly recommended as the preferred approach for choosing a DMT in MS.

Several approaches have been checked as the best method to the SDM process including text-, video- and web-based [23–26]. However, their reliability is not well demonstrated. Although the studies pointed out that the SDM process improved patient satisfaction and lower decisional conflict [26], all the evidence is centered in positive outcomes due to the use of decision aids, not in the perceived quality of the process itself. There are several elements of the decision-making process that can be measured, including the outcome of decision, readiness to plan, and decision quality [26].

Decision quality is known as the consistency of the individual's decision with their beliefs, satisfaction with the decision, participation in decision-making and patient-clinician communication [13]. Therefore, simply measuring decision outcomes is not a meaningful indicator of quality, as the eventual outcome can be dependent upon many external factors [24].

We developed a questionnaire based on two already validated scales, TSQM and DCS, looking to assess 4 important aspects of the decision quality: the satisfaction with the information during the SDM (items 01, 02, 03, 04, 05, 06, 11, 12 and 25), the adverse events during the treatment (items 15, 16, 17, 18 and 19), the convenience with the chosen-DMT (items

## Parallel Analysis

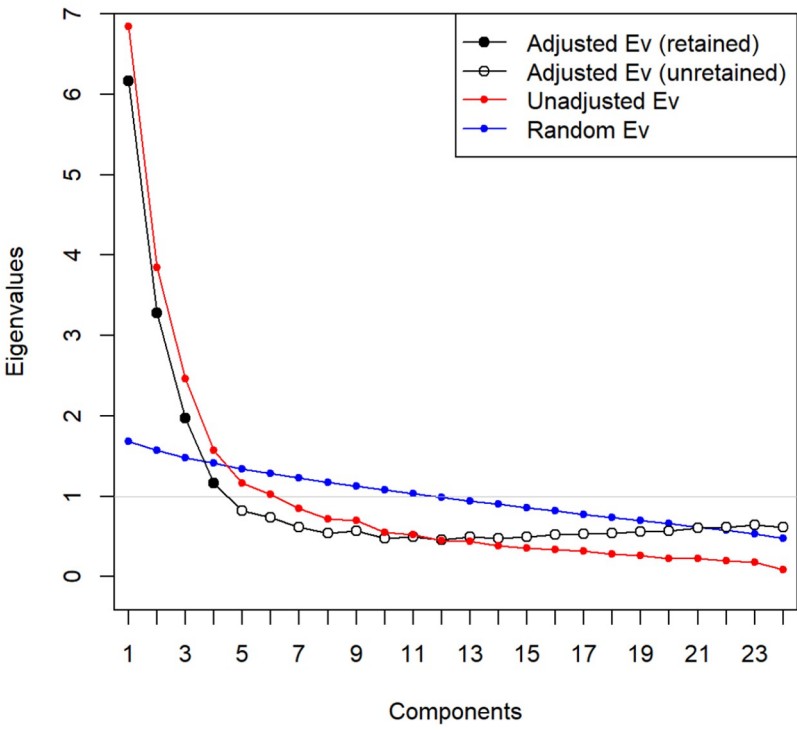

**Fig 1. Parallel analysis.**

7,8,10,20 and 21) and information reliability (items 13, 14, 22, 23 and 24). The exploratory factor analysis found the existence of these four dimensions in the AGA-25 scale. Moreover, this questionnaire has demonstrated a satisfactory internal reliability for all the factors, showing a Cronbach's coefficient higher or equal to 0.7 in all the subscales and a global coefficient of 0.88. This is the first scale, developed for MS patients, however, this scale could be validated for other chronic conditions.

Other questionnaires, such as the Control Preferences Scale or the 9-item Shared Decision-Making Questionnaire (SDM-Q-9) have been previously developed and validated in Spanish for different pathologies including MS [28, 29]; however, our scale is unique in the sense of evaluating satisfaction with the information obtained during the process itself and offers an improvement opportunity for physicians and in their relationship with patients.

In future work, test stability of the questionnaire will be assessed to ensure that, there is no temporal change in the responses and all patients understood correctly all the items. Moreover, a bigger sample of PwMS is been collected to probe the internal four structure of the survey through confirmatory factor analysis (CFA). Nevertheless, we are aware of limitations of this study due to lower rates of higher-activity DMT, which could lead to bias.

## Conclusion

This is the first questionnaire evaluating the quality of information given during the SDM process in multiple sclerosis patients. This questionnaire aims to determine if the chosen method for the SDM process is useful and well accepted in the outpatient clinic and offers an improvement opportunity.

**Table 4. Factorial loadings.**

|  | SatisfactionSDM | AdverseEffectsDMT | Information Reliability | DMTConvenience |
|---|---|---|---|---|
| Item 01 | **-0.17** | 0.00 | -0.02 | 0.10 |
| Item 02 | **-0.35** | 0.03 | -0.06 | 0.11 |
| Item 03 | **-0.37** | -0.02 | -0.06 | 0.02 |
| Item 04 | **-0.42** | -0.03 | -0.09 | -0.10 |
| Item 05 | **-0.23** | 0.02 | 0.03 | 0.05 |
| Item 06 | **-0.20** | -0.02 | 0.10 | -0.13 |
| Item 07 | -0.24 | 0.02 | 0.09 | **-0.31** |
| Item 08 | -0.31 | 0.00 | 0.16 | **-0.39** |
| Item 10 | -0.06 | -0.05 | 0.05 | **0.08** |
| Item 11 | **-0.37** | -0.01 | -0.05 | -0.04 |
| Item 12 | **-0.27** | 0.02 | 0.02 | 0.02 |
| Item 13 | 0.06 | 0.04 | **0.45** | -0.11 |
| Item 14 | 0.03 | 0.04 | **0.43** | -0.10 |
| Item 15 | -0.01 | **-0.29** | -0.02 | 0.03 |
| Item 16 | -0.02 | **-0.48** | 0.00 | -0.01 |
| Item 17 | 0.04 | **-0.53** | 0.02 | -0.02 |
| Item 18 | -0.00 | **-0.52** | -0.02 | -0.09 |
| Item 19 | 0.02 | **-0.35** | 0.03 | 0.08 |
| Item 20 | -0.17 | -0.02 | 0.04 | **0.60** |
| Item 21 | -0.13 | 0.02 | 0.09 | **0.48** |
| Item 22 | 0.03 | -0.00 | **0.43** | -0.03 |
| Item 23 | 0.01 | -0.04 | **0.41** | 0.13 |
| Item 24 | -0.02 | -0.05 | **0.39** | 0.16 |
| Item 25 | **-0.17** | -0.02 | 0.15 | 0.11 |

Our study demonstrated good reliability. This questionnaire evaluated 4 aspects of the SDM process and DMT satisfaction; each subscale demonstrated also acceptable reliability.

## Supporting information

**S1 File.**
(XLSX)

## Author Contributions

**Conceptualization:** Elena Álvarez-Rodríguez, Inés González-Suárez.

**Formal analysis:** María José Pérez-Haro.

**Investigation:** César Manuel Sánchez-Franco, Inés González-Suárez.

**Methodology:** Inés González-Suárez.

**Project administration:** César Manuel Sánchez-Franco, Laura Bello-Otero, Marta Aguado-Valcarcel.

**Supervision:** Inés González-Suárez.

**Validation:** Elena Álvarez-Rodríguez, Laura Bello-Otero, Marta Aguado-Valcarcel, Inés González-Suárez.

**Writing – original draft:** Inés González-Suárez.

**Writing – review & editing:** María José Pérez-Haro, Marta Aguado-Valcarcel, Inés González-Suárez.

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
