## [Decision Letter · Decision Letter 0]

25 Jan 2022

PONE-D-21-39578Development of a scale for the evaluation of the quality of the shared decision process in multiple sclerosis patients.PLOS ONE

Dear Dr. Suárez,

Thank you for submitting your manuscript to PLOS ONE. After careful consideration, we feel that it has merit but does not fully meet PLOS ONE’s publication criteria as it currently stands. Therefore, we invite you to submit a revised version of the manuscript that addresses the points raised during the review process.

We look forward to receiving your revised manuscript.

Kind regards,

Fatih Özden, PhD

Academic Editor

PLOS ONE

Journal Requirements:

a) Did participants provide their written or verbal informed consent to participate in this study?

3. We note you have included a table to which you do not refer in the text of your manuscript. Please ensure that you refer to Table 1 in your text; if accepted, production will need this reference to link the reader to the Table.

4. Please upload a copy of Supporting Information S1 File. XLS file.  which you refer to in your text on page 6.

Additional Editor Comments:

Dear Authors,

The reviewers now completed their reviews. Both three reviewers has indicated their good comments on the manuscript, however two of them requested minor revisions. Please carefully fulfill these comment, then submit it again. Please also provide a response to their comments with a separate form.

King Regards

Reviewers' comments:

Reviewer's Responses to Questions

**Comments to the Author**

1. Is the manuscript technically sound, and do the data support the conclusions?

Reviewer #1: Yes

Reviewer #2: Yes

Reviewer #3: Yes

2. Has the statistical analysis been performed appropriately and rigorously? 

Reviewer #1: Yes

Reviewer #2: Yes

Reviewer #3: Yes

3. Have the authors made all data underlying the findings in their manuscript fully available?

Reviewer #1: Yes

Reviewer #2: Yes

Reviewer #3: Yes

4. Is the manuscript presented in an intelligible fashion and written in standard English?

Reviewer #1: Yes

Reviewer #2: Yes

Reviewer #3: No

5. Review Comments to the Author

Reviewer #1: I have read your article and it is fantastic!. I have just minor or may be technical errors you need to address. Otherwise the article is good to go for me.

1) You mentioned that you used PCA to select the factor structure. Are there other clustering methods you could have used to achieve thesame/similar results?

2) You mentioned you used Leave-One Item out Cross-validation (i.e. you deleted one item out, and re-examine their alpha value), which may be biased as the model was trained/built using the entire data. To avoid this kind of bias, one will have to do a Leave-one-out cross-validation (Leave one person out of the model) and refit the model with that person included, and check whether or not the alpha value changes. If the change in the alpha value with and without this person is too drastic, then the model is unstable with respect to the selected items. In that case, one will need to redefine the factor structure or use limited factors. This is what I call model validation. Therefore, in addition to the leave-one item out approach, that evluates the consistency of the factor structure, I will like the authors to do leave-one out validation (leave-one person out of the model) and refit their model with the 4 identified scales. You can then plot the observed alpha versus the predicted alpha to see how well the model agrees. This will guarantee that an already validated scale is applicable to Spanish cohort. Unless otherwise, please provide concrete details why this sort of analysis cannot be done. I am statisfied with the current results, and the achieve value of alpha =0.88.

Reviewer #2: A sound manuscript with a very thorough methodological context on a very particular and extremely important subject. In addition it provides extremely detailed data on the aspects that were investigated and consequently a solid indication of the patient's perspective on matters that may influence his/her reality

Reviewer #3: In the manuscript, the authors validate a decision-making scale in the Spanish language.

New therapeutic options available for the treatment of MS –even if more effective and easier to administer– may pose increased risks of severe side effects. Taking this into account, involvement of PwMS in the treatment decision-making process becomes even more imperative.

The work seems very solid to me, although the number of patients recruited is small. Only in the discussion, I think it is necessary to mention other decision-making scales validated in Spanish, such as the Control Preference Scale, and explain the advantages of using the scale evaluated in this manuscript.

6. PLOS authors have the option to publish the peer review history of their article (what does this mean?). If published, this will include your full peer review and any attached files.

Reviewer #1: **Yes: **Valery Fuh-Ngwa

Reviewer #2: **Yes: **Dimitrios Kitsos

Reviewer #3: No

---

## [Author Response · Author response to Decision Letter 0]

3 Mar 2022

You mentioned that you used PCA to select the factor structure. Are there other

clustering methods you could have used to achieve the same/similar results?

Yes, indeed. We could have used non metric multidimensional scaling and cluster analysis for

ordinal data, for instance.

Non metric multidimensional scaling is a statistical methodology used to visualize high dimen-

sional and complicated data sets in few dimensions (preferably in two). The goodness of the fit is

made through the stress parameter. Stress could be described as a value showing the difference

between the distance in the reduced dimension in comparison to the complete dimensional space.

Stress values greater than 20 are considered as a poor dimension selection, values between 10

and 20, reasonable, between 5 and 10, good, between 2.5 and 5, excellent and between 0 and

2.5, perfect.

The principal goal of cluster analysis for ordinal data is to classify the observations in groups, in

the way that, the categorical observations in one group are most similar to each other and the

differences between the groups are the most different as possible. This technique can be used as

dimensional reduction as it is able to describe the hidden structure of the objects.

Nevertheless, to perform principal components analysis was for us a priority because this statis-

tical tool is able to rank the dimensions/components based on the amount of variance explained

by the data and we want to explain the maximum amount of information by the new selected

dimensions.

V ar(P C1) > V ar(P C2) > ... > V ar(P CN )

Where N are the number of original variables and PC are the principal components.

2. You mentioned you used Leave-One Item out Cross-validation (i.e. you deleted one

item out, and re-examine their alpha value), which may be biased as the model was

trained/built using the entire data. To avoid this kind of bias, one will have to do

a Leave-one-out cross-validation (Leave one person out of the model) and refit the

model with that person included, and check whether or not the alpha value changes.

If the change in the alpha value with and without this person is too drastic, then

the model is unstable with respect to the selected items. In that case, one will need

to redefine the factor structure or use limited factors. This is what I call model

validation. Therefore, in addition to the leave-one item out approach, that evluates

the consistency of the factor structure, I will like the authors to do leave-one out

validation (leave-one person out of the model) and refit their model with the 4

identified scales. You can then plot the observed alpha versus the predicted alpha

to see how well the model agrees. This will guarantee that an already validated scale

is applicable to Spanish cohort. Unless otherwise, please provide concrete details

why this sort of analysis cannot be done. I am statisfied with the current results,

and the achieve value of alpha =0.88.

Cronbach’s alpha coefficients using the whole sample N = 203 are presented in the table 1.

Where raw alpha is the alpha that we would obtain if we deleted the item itself. (Leave-One

Item out Cross-validation). The global value of raw alpha is 0.88.

And for each factor, using the whole sample:

• SatisfactionSDM

The overall Cronbach’s alpha is 0.874.

• AdverseEffectsDMT

The overall Cronbach’s alpha is 0.925.

InfoReliability

The overall Cronbach’s alpha is 0.873.

DMTConvenience

The overall Cronbach’s alpha is 0.625.

If we perform leave one (person) out cross validation, the resulted alpha’s for each subset (N-1)

are plotted in figure 1.

Where the solid lines are the expected alpha values and the dotted lines the observed for each

dimension, as well as for the overall alpha, when all the items are included.

As we can see the observed alpha values are quite stable and they are in agreement with the

expected values.

Reviewer #3 The work seems very solid to me, although the number of patients recruited is small.

Only in the discussion, I think it is necessary to mention other decision-making scales validated in

Spanish, such as the Control Preference Scale, and explain the advantages of using the scale evaluated

in this manuscript. Other scales are now mentioned and the main differences with prior scales have

been highlighted. Our scale is unique in evaluating satisfaction with the information obtained during

the process itself and offers an improvement opportunity for physicians and in their relationship with

patients.

---

## [Decision Letter · Decision Letter 1]

25 Apr 2022

Development of a scale for the evaluation of the quality of the shared decision process in multiple sclerosis patients.

PONE-D-21-39578R1

Dear Dr. Suárez,

We’re pleased to inform you that your manuscript has been judged scientifically suitable for publication and will be formally accepted for publication once it meets all outstanding technical requirements.

Kind regards,

Fatih Özden, PhD

Academic Editor

PLOS ONE

Additional Editor Comments (optional):

Reviewers' comments:

Reviewer's Responses to Questions

**Comments to the Author**

1. If the authors have adequately addressed your comments raised in a previous round of review and you feel that this manuscript is now acceptable for publication, you may indicate that here to bypass the “Comments to the Author” section, enter your conflict of interest statement in the “Confidential to Editor” section, and submit your "Accept" recommendation.

Reviewer #1: All comments have been addressed

Reviewer #4: All comments have been addressed

2. Is the manuscript technically sound, and do the data support the conclusions?

Reviewer #1: Yes

Reviewer #4: Yes

3. Has the statistical analysis been performed appropriately and rigorously? 

Reviewer #1: Yes

Reviewer #4: Yes

4. Have the authors made all data underlying the findings in their manuscript fully available?

Reviewer #1: Yes

Reviewer #4: Yes

5. Is the manuscript presented in an intelligible fashion and written in standard English?

Reviewer #1: Yes

Reviewer #4: Yes

6. Review Comments to the Author

Reviewer #1: Excellent work! I am very statisfied with the current results following the Leave-one-person Out cross-validation. Both the observed and expected alpha seems very reasonable to me. Great job there!

Reviewer #4: Authors well answered to all comments. Statistical analyses wer well performed and the paper was clear and well written.

7. PLOS authors have the option to publish the peer review history of their article (what does this mean?). If published, this will include your full peer review and any attached files.

Reviewer #1: **Yes: **Valery Fuh-Ngwa

Reviewer #4: No

---

## [Editor Report · Acceptance letter]

6 May 2022

PONE-D-21-39578R1 

Development of a scale for the evaluation of the quality of the shared decision process in multiple sclerosis patients. 

Dear Dr. Suárez:

I'm pleased to inform you that your manuscript has been deemed suitable for publication in PLOS ONE. Congratulations! Your manuscript is now with our production department. 

Kind regards, 

on behalf of

Dr. Fatih Özden 

Academic Editor

PLOS ONE